# Right Pulmonary Artery Distensibility Index in Heartworm Infected Dogs: Are the Different Methods Leading to Same Results?

**DOI:** 10.3390/ani13030418

**Published:** 2023-01-26

**Authors:** Angelo Basile, Ettore Napoli, Emanuele Brianti, Luigi Venco

**Affiliations:** 1Centro Nefrologico Veterinario, 95123 Catania, Italy; 2Department of Veterinary Sciences, University of Messina, 98168 Messina, Italy; 3Ospedale Veterinario Città di Pavia, 27100 Pavia, Italy

**Keywords:** *Dirofilaria immitis*, dog, right pulmonary artery distensibility index, echocardiography

## Abstract

**Simple Summary:**

*Dirofilaria immitis* infection in dogs induces damage to pulmonary arteries, which in turn, if severe enough, can lead to pulmonary hypertension and clinical signs of right-sided congestive heart failure. The initial vascular damage results in progressive reduction of the elastic properties of the vessel wall and consequent loss of its distensibility. Therefore, the reduced distensibility of the pulmonary artery wall becomes an early indicator of pulmonary hypertension. Pulmonary artery distensibility can be measured at the level of the right pulmonary artery, using an echocardiographic technique, obtaining the so-called distensibility index. Several studies on the distensibility index have been reported in the veterinary literature, where different methods of measurements have been used. This study was conducted to determine whether the results obtained with the different methods are superimposable in terms of clinical information. The present work indicates that, although there is a statistical agreement between the results of the different tests, they cannot be used interchangeably from a clinical point of view.

**Abstract:**

Canine Heartworm Disease (HWD) is mainly a pulmonary vascular disease. The reduction of the Pulmonary Artery Distensibility (PAD) is an early index of pulmonary vascular disease. Echocardiographic evaluation of the Right Pulmonary Artery Distensibility index (RPADi) is calculated as the percentage change in diameter of the right pulmonary artery (RPA) between systole and diastole. Historically, two main methods have been used for RPADi calculation: The Venco method and Visser method; however, different hybrid methods have also been used by other authors. Therefore, it could be difficult for a clinician to decide which method to apply and how to interpret the results based on the reference values reported. The aim of this study was to compare the RPADi obtained by five different techniques (Venco classic, Venco modified, Visser classic, Visser modified 1, and Visser modified 2). The study design was a retrospective, single center, observational study. Forty-seven client-owned dogs were included. The measurements were performed off-line as an average of three consecutive cardiac cycles by a single investigator blinded to the dogs’ diagnosis. The RPADi was satisfactorily obtained by all methods in all dogs. Intra-observer measurement variability was clinically acceptable both for systolic and diastolic measurements. Although the Bland–Altman test showed a statistical agreement between the various methods used to calculate the RPADi, these methods cannot be used interchangeably in a clinical setting. Instead, the measurement method and reference values should always be specified.

## 1. Introduction

Canine Heartworm disease, caused by *Dirofilaria immitis*, is mainly a pulmonary vascular disease, which, in advanced stages, may affect the right heart because of precapillary pulmonary hypertension (PH), pulmonary thromboembolism, and/or direct worm invasion of the right chambers [1]. Pulmonary vessel alterations due to heartworms result in proliferative endarteritis, vascular remodeling, and thromboembolism, causing a sustained elevation of pulmonary arterial pressures [2,3]. Damage to peripheral arteries and arterioles begins soon after the onset of infestation. This is due to the multiplication and migration of smooth muscle cells to the intima, which creates a villous proliferation in the inner surface of the artery. Consequently, there is thickening of the arterial walls, which in addition to the rough texture of the intimal surface, cause a narrowing of the vessel lumen, especially in the small branches [1,2,3]. In the pulmonary circulation, small vessels contribute to about 80% of pulmonary arterial compliance (PAC), a function closely linked to the elasticity of the vessel wall and defined as the relative change in volume for each change in pressure. As lesions develop, the entire pulmonary vascular tree becomes stiffer and loses compliance, so there is an increase in pulmonary arterial pressure that leads to a chronic development of pulmonary hypertension (PH). Furthermore, the death of parasites triggers pulmonary thromboembolism, which also causes acute episodes of PH and contributes to the chronic development of PH [4,5].

An important aspect of the pathophysiology of pulmonary vascular lesions, related to heartworm disease, is represented by the increase in mean pulmonary artery pressure caused by peripheral vascular disease. This, in turn, induces a reduction in PAC and consequently in Pulmonary Artery Distensibility (PAD) which is defined as a relative change in vessel diameter for a given change in pressure, but simultaneously increases stiffness (reciprocal of compliance) [6,7]. It should be emphasized that the reduction of distensibility can be detected in the earliest stages of vascular disease, in the absence of significant and irreversible damage to the vessel wall [7,8]. Furthermore, considering the inverse proportionality between pulmonary vascular resistance (PVR) and PAC, with a hyperbolic ratio [9], interestingly, in the pulmonary circulation, contrary to the systemic circulation, PAC (and therefore the PAD) and PVR are uniformly distributed throughout the vascular bed. Hence, PAC and PAD change significantly and sooner, even before the increase of PVR measured by invasive methods, making stiffness and PAD early markers of pulmonary vascular disease [10].

Based on these statements, it appears reasonable that the effects of peripheral vascular disease induced by HWD could be detected early at the level of proximal pulmonary vessels (main pulmonary trunk, left and right pulmonary artery). Therefore, reduction of Pulmonary Arterial Distensibility (PAD) becomes an early index of pulmonary vasculopathy which can be detected even before other findings consistent with PH (see Figure 1).

Non-invasive imaging modalities as Magnetic Resonance Imaging (MRI) [11,12,13,14] and echocardiography [15,16] have been used to measure PAD. Echocardiographic assessment of the right pulmonary artery distensibility index (RPADi) was described as a valuable method for evaluation of PAD in dogs [17,18,19,20,21,22,23,24,25] (Table 1). The rationale for using the right pulmonary artery (RPA) to assess PAD by echocardiography, is represented by its easy evaluation with this diagnostic tool. Furthermore, a previous study utilizing Computed Tomography Angiography (CTA) demonstrated that RPA distensibility yielded the best diagnostic value for PH in people [26]. The RPADi is calculated as the percentage change in diameter of the RPA between systole and diastole.

There are two main different published methods for RPADi calculation: the “Venco” method, in which the right pulmonary artery diameter is measured from leading edge-to-leading edge (le-le) using M-mode echocardiography from the right parasternal long axis view (RPLA) [17]; and the “Visser” method, which uses measurements of the right pulmonary artery from trailing edge-to-leading edge (te-le) using B-mode echocardiography from the right short axis view (RPSA) [19]. Moreover, hybrid methods have also been used [22,24,25,26,27]. Table 1 summarizes the published studies.

In view of so many different methods of assessment of RPADi, it has become difficult for a routine clinical use to decide which method applies best and how to compare the results of studies where different methods were used. Therefore, an obvious question arises: do the different methods lead to the same results?

The aim of this study was to compare the RPADi values obtained with five different techniques and to verify whether for routine clinical use all these methods can be used interchangeably. 

## 2. Materials and Methods

This was a retrospective, single-center, observational study. All animals included in the study were selected from the database of a previous epidemiological survey which sought to explore the distribution of canine heartworm disease in the Island of Linosa (data forthcoming). Client-owned dogs with different clinical presentations, both healthy and naturally infected with *D. immitis*, were included. Echocardiographic examinations were performed by a single investigator (A.B.) at different time periods, as decided in the study protocol. The dogs were not sedated and were gently restrained by the owner in right and left lateral recumbences on a dedicated table for the echocardiographic examination. For each patient, cineloops of five consecutive cardiac cycles were acquired according to the recommendations for standard transthoracic echocardiography [28,29], in order to acquire a minimum data set for evaluation of cardiac chamber dimensions, wall thicknesses, structure and function of all cardiac valves, right and left ventricular systolic and diastolic function, presence of any pericardial fluid, presence of pulmonary hypertension, and to perform the Speckle Tracking echocardiographic study for the right ventricle. In addition, the following off-axis view were performed: (a) RPLA, optimized for the roof of the left atrium, zoom view, to visualize the right pulmonary vein and RPA (View 1) (Figure 2a). An M-mode study was performed on this view with the interrogation cursor aligned in the middle of the RPA and perpendicular to the vessel wall (Figure 2b). The end-systolic dimension of the RPA was measured at the maximum diameter (usually at the T wave) and the end-diastolic diameter was measured at its smallest dimension (at the Q wave Figure 2b); (b) RPSA, optimized for RPA, to visualize its longest path (View 2) (Figure 2c). An M-mode study was also performed on this view, with cursor positioned approximately at the midpoint between the bifurcation of the main pulmonary trunk and the distal end of the RPA (Figure 2c).

The machine used was the GE Vivid Q premium (General Electric, Vivid iq Premium, version 1.2.8) equipped with 1.5–4.6 MHz phased-array transducer (M5ScRS) and 2.7–8 MHz phased-array transducer (6SRS). The M-mode echocardiographic examination was acquired at a speed of 100 mm/s and/or included at least three cycles in each single image. All echocardiographic examinations acquired from the study population were saved on a dedicated archiving system (echoPAC, version 204, General Electric).

For the calculation of the RPADi, only M-mode studies of good quality and, with a clearly visible ECG trace (recognizable P, QRS, and T waves) from both View 1 and View 2 images were selected by the echoPAC. The measurements were made anonymously, in order to blind the investigator to the original assessment. The measurements were performed by a single investigator (A.B.) in randomized order and calculated as an average of three consecutive cardiac cycles.

The RPADi was calculated with the following formula: [(RPAs-RPAd)/RPAs]×100, where the RPADi value is expressed as a percentage and RPAs and RPAd represent the systolic (maximum) and diastolic (minimum) diameters of the RPA, respectively.

The RPADi was calculated using five methods:(1)method 1 (“Venco classic”): obtained by M-mode study using View 1. The diameters of the RPA were measured with “the leading edge to leading edge” (le-le) measurement convention [17,28] (Figure 3).(2)method 2 (“Venco modified”): obtained in the same way as for method 1, but diameters of the RPA were measured with “the trailing edge to leading edge” (te-le) measurement convention (Figure 3).(3)method 3 (“Visser classic”) [19]: obtained by B-mode study using View 2. Minimum diastolic and maximum systolic internal diameters were measured with “the trailing edge to leading edge” (te-le) method, using the same location of the RPA. The measurements were performed as perpendicular as possible to the internal borders, where RPA was clearly visible, starting from the frame with the largest diameter, and subsequently, scrolling frame-by-frame, the smallest diameter of RPA was measured, attempting to measure it at the same location.(4)Method 4 (“Visser modified 1”, Visser “te-le”): using an M-mode image obtained from View 2, the diameters are measured using the “trailing edge to leading edge” (te-le) method.(5)Method 5 (“Visser modified 2”, Visser “le-le”): using an M-mode image obtained from View 2, the diameters are measured using the “leading edge to leading edge” (le-le) method.

In all methods, the systolic dimension of the pulmonary artery was measured at the maximum diameter (usually at the T wave) and diastolic diameter at its smallest dimension (at the Q wave). 

### Statistical Analysis

Statistical analyses were performed using a free software package Jamovi (the jamovi project, version 1.6) and a commercial software MedCalc (MedCalc, version 18.2.1). 

Descriptive statistics were generated and normality testing was assessed both graphically and with the Shapiro–Wilk test. Normality was rejected for *p* ≤ 0.05 for Shapiro–Wilk or if te coefficient of skewness or coefficient of kurtosis had *p* ≤ 0.05.

For each method, the values of the arithmetic mean, standard deviation (normal data), median, and interquartile range (for those not normal) were calculated from the values of the three measurements of systolic and diastolic diameters. The coefficient of variation was also calculated.

Linear regression analysis was first performed in order to investigate the relationship between the “Venco” classic method and the others.

The agreement among RPADi measurements with the different methods were evaluated for bias and limits of agreement (LoAs) by the Bland–Altman method. “Venco” classic was used as the reference method for the other four methods, as it was historically the first study published. Additionally, the authors provide RPADi values “associated” with direct measurements of pulmonary artery pressure. Furthermore, regarding the RPADi topic, this study’s results are currently most cited in the literature.

In order to identify fixed bias, one-sample *t*-tests comparing the difference between Venco classic and the other methods were performed. A fixed bias is indicated if the difference differs significantly from 0 (*p*-value ≤ 0.05) or, equivalently, if the 95% Confidence Interval (CI) for the differences does not include 0 [30].

Proportional bias was investigated by linear regression of the differences between Venco classic and the other methods. Proportional bias does exist if the slope *b* in the linear equation y = *a* + *b*x differs significantly from 0 (i.e., *p*-value ≤ 0.05) [30].

To verify echocardiographic measurements variability and thus repeatability, intra-observer measurement variability was quantified by the average coefficient of variation (CV) over three consecutive cardiac cycles, using the following equation:CV = (mean difference between measurements/mean of measurements) × 100, expressed as a percentage. 

The average percent coefficient of variation (CV) was used to quantify echocardiographic measurement variability studies, using the following equation:CV = (standard deviation of the measurements/average of the measurements) × 100. 

## 3. Results

Forty-six echocardiographic examinations satisfied the inclusion criteria. RPADi values were satisfactorily obtained by all methods. 

The results of linear regression analysis are summarized in Figure 4.

The results obtained by Bland–Altman analysis are summarized in Table 2 and Figure 5.

No fixed bias (Table 3) or proportional bias (Table 4) were founded by one-sample *t*-test of differences and linear regression of differences against means, respectively. These two conditions make the Bland–Altman test able to be used safely in comparison between the methods and determines if they may be used interchangeably in clinical practice.

Analyzing the three statistics, bias (mean differences), precision, and limits of agreement, which are deducible from the graphs and useful for evaluating the performance of the alternative methods compared to Venco classic [31], we can observe that Venco te-le, Visser classic, and Visser te-le measure values of RPADi 1.6% higher than the Venco classic, while Visser le-le underestimates by an average of 1.8%

Regarding the precision of the methods (obtained from the values of the standard deviation of the differences between pairs of measured values of each single method compared to the Venco classic), the results showed this to be equal to 4.6% for the Venco te-le, 8.32% for the Visser classic, and 9.04 and 8.97, respectively, for Visser le-le and Visser te-le.

The limits of agreement, calculated as bias ± 1.96 standard deviation, are described in Table 5, along with the 95% CI of the minimum and maximum values.

Furthermore, by visually analyzing the distribution of the measurements within the limits of agreement, Venco te-le showed a good agreement with Venco classic, with a random distribution of the points around the bias line and inside the 95% limits of agreement. Instead, Visser classic showed a subtle trend to overestimate the lower values and underestimate the upper ones. This trend becomes more marked for both M-mode-modified Visser methods.

Moreover, if we examine standard deviation and the lower and upper 95% limits of agreement values, the statistical agreement shown by the Bland–Altman plots is not clinically useful, considering what each RPADi value means in term of pulmonary pressure; just a 10 % difference between methods could represent a shift from one pressure class to another.

The data relating to the Intra-observer measurement variability are summarized in Table 3. They have been shown to be clinically acceptable for all methods with a CV always below 6%.

## 4. Discussion

The present study has shown that, although the Bland–Altman test has highlighted a statistical agreement between the different tests, they cannot be used interchangeably from a clinical point of view, considering the standard deviations of the differences of between 5 and 10% compared to the "classic" Venco method.

Of the five tested methods, four have already been described in previous studies: Venco “classic” [17,20,21,23], Venco “te-le” [22,23,24,25], Visser “classic” [19], and Visser “le-le” [24]. However, different methodologies were used in these studies. First, different types of PH were included: only pre-capillary in the studies by Venco, Serrano, Falcon-Cordon, Roels, and Maerz [17,20,23,24] versus a mixed group of pre-capillary and post-capillary in the studies by Visser and Chan [19,21]. In addition, in the studies by Venco [17], Serrano [20], Falcon-Cordon [23], and Maerz [24] the sample consisted only of patients affected by HWD. Second, the diagnosis of PH was performed using the tricuspid regurgitation (TR) velocity, which represents an estimated, but not measured value of pulmonary pressure. Distinctly, in Venco’s study [17], the RPADi values were also compared with values measured by invasive pulmonary artery catheterization, although this was performed only for the group of dogs undergoing surgical heartworm removal due to the high worm burden. Third, the definition of PH and its severity grading (mild, moderate, and severe) is different in the various studies: the normal cutoff value is estimated to be 36 mmHg in Visser and Chan, and 30 mmHg in Venco [17], Serrano [20], Falcon-Cordon [23], Roels [22], and Maerz [24]. Finally, the normal value of RPADi which distinguishes "normal" subjects from those affected by "mild PH" is considered < 29% by Venco [17], Serrano [20], Falcon-Cordon [23], Roels [21], and Maers [24], while Visser indicates a RPADi cutoff of 34.6% to predict a TR of 36 mmHg (by means of ROC curve and Youden index).

The echocardiographic method employed for the measurement of RPADi was the M-mode in all published studies, except for Visser’s study [19,20,21,22,23,24,25]. Theoretically, the M-mode method could be superior in the evaluation of RPADi based on a more precise delineation of the point where the diameter of the pulmonary artery should be measured in systole and diastole, and for the better temporal resolution compared to the B-mode.

The RPLA view (used by Venco Serrano, Chan and Falcon-Cordon) [17,20,21,23,25], optimized for the roof of the left atrium, shows the right pulmonary artery in transverse section, allowing study of its diameter in a unique and repeatable point. It is the author’s opinion (AB) that, improving the echocardiographic method by framing the group of pulmonary artery and vein as centrally as possible and acquiring the clips in zoom-viewed mode will allow a further increase in the temporal and spatial resolution of the M-mode.

The RPSA view shows the right pulmonary artery in longitudinal section, leaving wide discretion to the operator to decide the point of artery where to perform the measurements each time. Furthermore, there are no reported studies that compare the diameter values of the right pulmonary artery, scanned in longitudinal view, in different points.

Another difficulty in the comparison of the different published studies as in Falcon-Cordon [23] and Chan [21], is that the authors perform the measurements with the "classic" Venco method despite using the data of Visser’s study for both positivity cut-offs and reference intervals. Furthermore, Maerz et al. and Birettoni et al. [24,25] apply a "hybrid" method of measurement (RPSA, M-mode, and le-le) using Venco reference values.

Several limitations of this study should be acknowledged owing first to its single center and retrospective nature.

Second, all the echocardiographic examinations and measurements were performed by a single investigator (A.B.), although, the exams were made anonymous after the acquisition to blind the measurement from the original analysis. Another limit can be considered having considered Venco’s method [16] as the reference method. However, this operational choice has been already justified above.

In summary, RPADi, regardless of the calculation method used, represents a very useful and easily obtainable echocardiographic marker of PH and for this reason, in recent years, it has gained a lot of attention both among veterinary cardiologists and veterinary parasitologists involved in studies of HWD. Measuring RPADi, at an easily accessible site such as the proximal pulmonary artery, permits us to evaluate the overall effects of events which are mainly affecting the peripheral pulmonary circulation. This has been shown helpful in early diagnosis of PH, even in the absence of other echocardiographic signs, such as tricuspid and pulmonary regurgitation and requires a very short operator learning curve.

Further studies are mandatory to determine if RPADi can be used as a single PH marker or if it needs to be included in a multiparametric diagnostic approach. Another point to clarify in future studies is if different reference ranges are needed in cases of pre-capillary, post-capillary, and/or combined forms of PH. Finally, a multicentric study would be necessary to establish if other variables, such as breed or age could influence RPADi reference intervals.

## 5. Conclusions

In conclusion, the present retrospective single-center study indicates that different methods for calculating RPADi are not interchangeable, and that the measurement method and reference values should always be specified.

## Figures and Tables

**Figure 1 animals-13-00418-f001:**
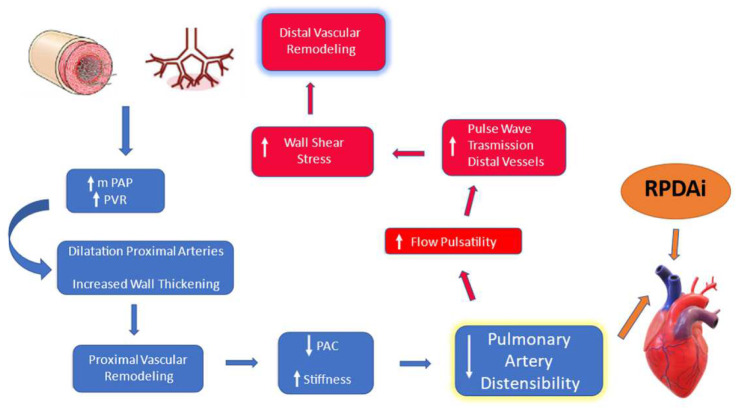
Schematic representation of the interrelationship loop between peripheral vascular disease and modification of pulmonary distensibility. mPAP, mean pulmonary artery pressure; PVR, pulmonary vascular resistance; PAC, pulmonary arterial compliance.

**Figure 2 animals-13-00418-f002:**
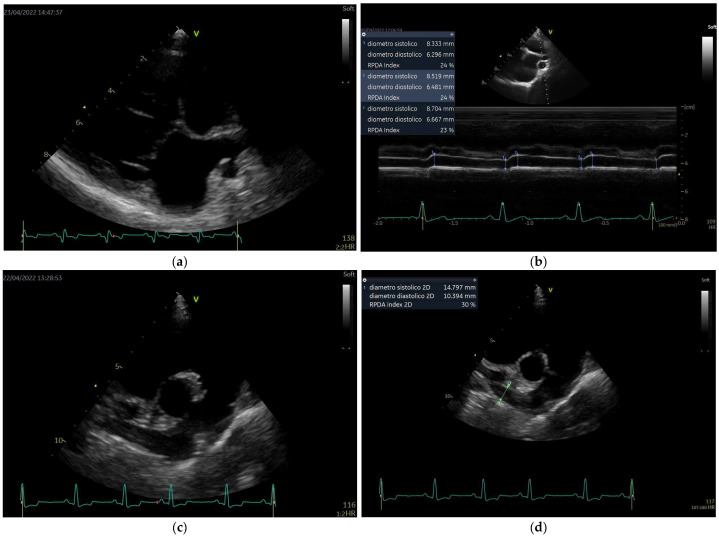
Echocardiographic views and measurement method. (**a**): View 1 (see full text); (**b**): diameter measurement of right pulmonary artery in M-mode from View 1; (**c**): View 2; (**d**): diameter measurement of right pulmonary artery in B-mode from View 2.

**Figure 3 animals-13-00418-f003:**
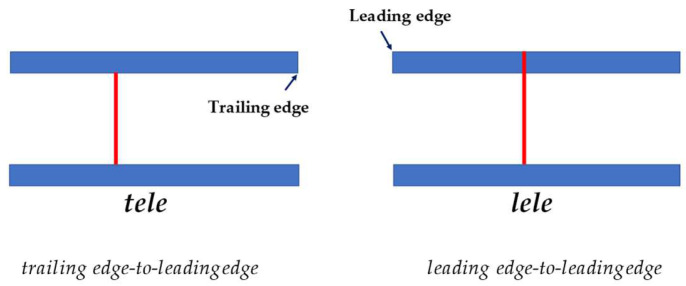
A schematic representation of two measurement methods used to determine right pulmonary artery diameter at each measurement site. tele, trailing edge-to-leading edge; lele, leading edge-to-leading edge.

**Figure 4 animals-13-00418-f004:**
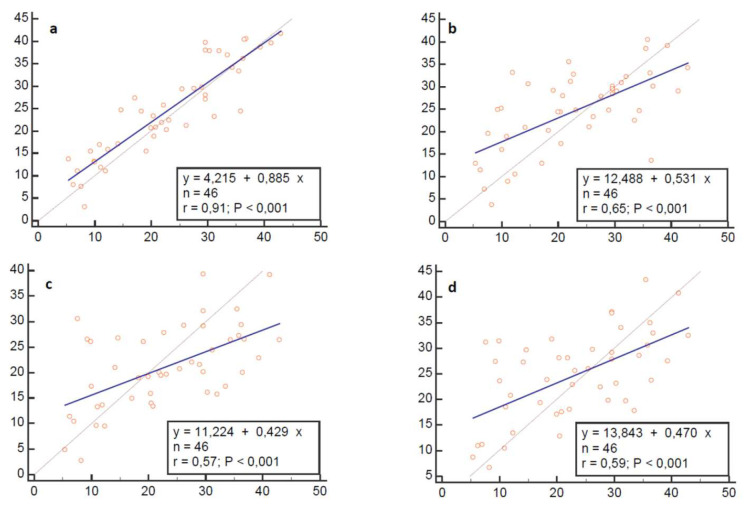
Linear regression analysis plots. Regression model: Alternative method (y) = a + b (Venco classic). Blue line represents line of equality (y = x). (**a**), Venco te-le; (**b**), Visser classic; (**c**), Visser le-le; (**d**), Visser te-le.

**Figure 5 animals-13-00418-f005:**
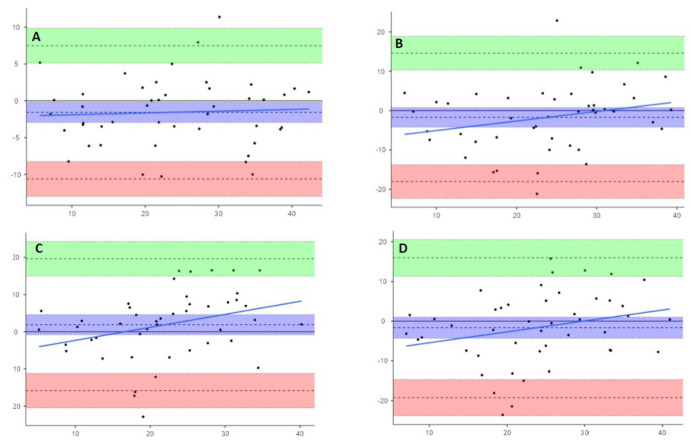
Bland–Altman plots showing paired difference between Venco classic and (**A**), Venco te-le; (**B**), Visser classic; (**C**), Visser le-le; and (**D**), Visser te-le. Upper and lower dashed lines show LoAs. Central dashed lines show the means of the differences. Light blue line is the proportional bias line. Light green and rose areas are the 95% Confidence Intervals of upper and lower LoAs. Purple area represents the 95% Confidence Interval of the mean of differences.

**Table 1 animals-13-00418-t001:** RPADi calculation methods published in the veterinary literature. Bold type indicates papers applying the “Venco” method. RPLA, Right Parasternal Long Axis; RPSA, Right Parasternal Short Axis.

Author	Year	Echocardiographic Method	View	Measuring Method
**Venco**	**2014**	**M-mode**	**RPLA**	**lele**
Vezzosi	2015	M-mode	RPSA RPLA	Not Specified
Visser	2016	B-mode	RPSA	tele
Birettoni	2016	M-mode	RPLA RPSA	tele
**Serrano**	**2017**	**M-mode**	**RPLA**	**lele**
**Chan**	**2019**	**M-mode**	**RPLA**	**lele**
Roels	2019	M-mode	RPLA	tele
**Falcon-Cordon**	**2019**	**M-mode**	**RPLA**	**lele**
Maerz	2020	M-mode	RPSA	lele

**Table 2 animals-13-00418-t002:** Results of Multiple Variable Bland–Altman comparison.

Reference Method	Venco Classic				
Systematic Differences					
Variable	*n*	Differences			
		Mean	SD	95%CI	
Visser classic	46	1.72	8.3	−0.755 to 4.2	
Venco te-le	46	1.57	4.6	0.19 to 2.93	
Visser le-le	46	−1.89	9.04	−4.57 to 0.79	
Visser te-le	46	1.66	9	−1.01 to 4.32	
Limits of Agreement					
Variable	*n*	Limits of Agreement			
		Lower limit	95% CI	Upper limit	95% CI
Visser classic	46	−14.6	−18.9 to −10.3	18	13.7 to 22.2
Venco te-le	46	−7.5	−9.8 to −5.1	10.6	8.2 to 1.9
Visser le-le	46	−19.6	−24.2 to −15	15.8	11.2 to 20.5
Visser te-le	46	−1.9	−20.5 to −11.3	19.2	14.7 to 23.8
Regression					
Variable	Intercept	95% CI	Slope	95% CI	*p*
Visser classic	12.4	7.7 to 17.2	−0.46	−0.65 to −0.27	<0.0001
Venco te-le	4.2	1.01 to 7.4	−0.11	−0.24 to 0.01	0.0726
Visser le-le	11.2	6.4 to 16	−0.57	−0.76 to 0.38	<0.0001
Visser te-le	13.8	8.8 to 18.8	−0.53	−0.72 to −0.33	<0.0001
Absolute Percentage Error					
Variable	*n*	Median	95% CI	95th Percentile	
Visser classic	46	19.93%	14.36 to 34.6	15.4%	
Venco te-le	46	14.64%	8.4 to 24.78	66.6%	
Visser le-le	46	23.72%	19.75 to 36.14	168.01%	
Visser te-le	46	24.58%	16.5 to 35.4	199.27%	

**Table 3 animals-13-00418-t003:** Outcomes of analysis of differences by one-sample *t*-test for fixed bias.

Differences	Mean Difference (± SEM)	95% CI for Mean Difference	*t*	*p*-Value
Venco te-le	−1.56 ± 0.68	−2.94 to −0.19	−2.3	0.0261
Visser classic	−1.72 ± 1.22	−4.19 to 0.75	−1.4	0.1686
Visser le-le	1.89 ± 1.33	−0.80 to 4.57	1.4	0.1637
Visser te-le	−1.66 ± 1.32	−4.19 to 1.58	−1.2	0.215

SEM, standard error of the mean; 95% CI for mean difference, 95% confidence internal for mean difference; t, one-sample *t* statistic at d.f. = 45; *p*-value, two-sided *p* value from t-test.

**Table 4 animals-13-00418-t004:** Outcomes of Differences against means by linear regression analysis for proportional bias.

Regression	r	a	b	*p*-Value
Venco te-le	0.05	2.13	0.023	0.725
Visser classic	0.26	7.502	0.243	0.084
Visser le-le	0.32	5.83	0.35	0.0285
Visser te-le	0.26	8.235	0.276	0.076

r, regression coefficient; a and b, intercept and slope of the linear equation y= a + bx, respectively.

**Table 5 animals-13-00418-t005:** Coefficients of variation. Data expressed as mean and standard deviation (SD).

METODO		Mean	SD
**VENCO** **CLASSIC**	Systolic	4.1	3.1
	Diastolic	4.8	3.4
**VENCO** **tele**	Systolic	3.3	2.5
	Diastolic	5.7	4
**VISSER** **CLASSIC**	Systolic	3.7	2
	Diastolic	4.5	3
**VISSER** **tele**	Systolic	4.1	2.5
	Diastolic	4.9	3
**VISSER** **lele**	Systolic	3.6	2.5
	Diastolic	4.8	2.7

## Data Availability

Data may be obtained by contacting the corresponding author.

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
