# Peer review of "Right Pulmonary Artery Distensibility Index in Heartworm Infected Dogs: Are the Different Methods Leading to Same Results?"

_animals, 2023, doi:10.3390/ani13030418_

Round 1

Reviewer 1 Report

My recommendations follow.

Thanks

Author Response

Reviewer 1

A: We thank the Reviewer for her/his time and for her/his opinion of the manuscriscipt.

Introduction:

Abbreviations: The author needs to put the meaning of PVR. (page 2, line: 70).

A: done

Two arrows (white) are outside of figure 1 I recommend putting the initials in the description of figure 1 (page 3, line: 83).

A: done

Materials and Methods

Methods used to assess validity. I very much appreciate that the authors a Bland-Altman plot was constructed. However, because fixed and proportional biases cannot be determined independently from these plots, I recommend using ordinary least products regression analysis was used. (page 6, line: 203).

As regards the so-called 95% limits of the agreement it is clear from the foregoing that only when there is no proportional bias can this technique be used safely to decide whether one method corresponds well enough to the other so that either may be used in clinical practice. I recommend this article doi: 10.1046/j.1440-1681.2002.03686.x.

I would like to thank the reviewer in helping me to discover the papers John Ludbrook have written about the methods of agreement.

As you have suggested me, I reinforce the statistical section of the paper, starting with a linear regression between the reference method (Venco classic) and the others. Then in order to look for fixed bias I had runned a one-sample t-test of the differences of the methods and linear regression of differences between every single other method and Venco classic against the mean. I hope all these new data have satisfied your requests.

I very much appreciate that the authors, to verify the variability of the echocardiographic measurements, measured the variability of the intra-observer measurements. However, it would be interesting to add to inter-observer measures. (page 7, line: 209).

A: We thanks the reviewer for her/his comment, however the echocardiography was performed by a single operator (AB) that although the operator is highly trained this represent a limit of the study as already reported in line (page 10, lines: 297-299).

Results The author needs to fix figure 4. the name is on the graphic and not on the side.

A: done

Reviewer 2 Report

Authors have compared a useful echocardiographic index, the right pulmonary artery distensibility index, obtained by different methods in dogs. The manuscript is well written, the aim is well run, and the results brings noteworthy information to the field of cardiology. This Reviewer recommends the publication of this manuscript. 

Only a suggestion for the Authors:

There is another paper in veterinary literature about right pulmonary artery distensibility index in the dog: Birettoni F et al. Canine pulmonary vein-to-pulmonary artery ratio: echocardiographic technique and reference intervals. J Vet Cardiol. 2016 Dec;18(4):326-335. doi: 10.1016/j.jvc.2016.07.004. Authors should add it in the manuscript, please.

Author Response

Reviewer 2

Authors have compared a useful echocardiographic index, the right pulmonary artery distensibility index, obtained by different methods in dogs. The manuscript is well written, the aim is well run, and the results brings noteworthy information to the field of cardiology. This Reviewer recommends the publication of this manuscript. 

Only a suggestion for the Authors:

There is another paper in veterinary literature about right pulmonary artery distensibility index in the dog: Birettoni F et al. Canine pulmonary vein-to-pulmonary artery ratio: echocardiographic technique and reference intervals. J Vet Cardiol. 2016 Dec;18(4):326-335. doi: 10.1016/j.jvc.2016.07.004. Authors should add it in the manuscript, please.

A: We thanks the reviewer for her/his comment, the suggested reference has been added in the revised version of the manuscript and References numeration has been updated

Reviewer 3 Report

Dear authors,

i personally found the paper very interesting and useful for veterinary practitioners.

I have few question and comments.

line 121-122: data regarding size, age and breed of the dogs included are missing

line 234-235: there is mention to table x but i could not find it in the paper or in the unpublished material

line 248: the concept of clinically acceptable has been considered based on specific literature or the clinical practice? please clarify this point

Based on the current knowledge and on the author experience, suggestion regarding the specific tests to be applied could be provided to veterinary practitioners in the daily practice?

Author Response

Reviewer 3

Dear authors,

I personally found the paper very interesting and useful for veterinary practitioners.

I have few question and comments.

line 121-122: data regarding size, age and breed of the dogs included are missing

A: We thanks the reviewer for her/his comment, however, in our opinion data on size, age and breed of the dogs included are not fundamental for the Manuscript since they could not impair the study results

line 234-235: there is mention to table x but i could not find it in the paper or in the unpublished material

A: We thanks the reviewer for her/his comment it was just a typo, the table was table 3,

line 248: the concept of clinically acceptable has been considered based on specific literature or the clinical practice? please clarify this point

A: We thanks the reviewer for her/his comment the concept of clinically acceptable has been considered based on clinical practice of highly trained operators (AB and LV).

Based on the current knowledge and on the author experience, suggestion regarding the specific tests to be applied could be provided to veterinary practitioners in the daily practice?

Bland-Altman test showed a statistical agreement between the various methods used to calculate RPDAi, these methods cannot be used interchangeably in a clinical setting, and the suggestion is to always specify in the echocardiographic report the measurement method and reference values used.